# PICK YOUR TEXTUAL GRADIENTS

## ABSTRACT

Automated prompt optimization using textual gradients is a promising approach to improve the performance of Large Language Models (LLMs) with the guidance of natural language feedback. However, the iterative application of these gradients is notoriously unstable. We identify two primary sources of this instability: 1) gradient noise from correctly handled examples, and 2) a loss of generalization, where performance on simpler tasks declines due to over-specialization on complex cases. To address this, we propose a novel framework that stabilizes the optimization process through two core mechanisms: **Error-Driven Refinement** and **Regularized Verification**. First, the error-driven approach ensures a high-quality learning signal by exclusively generating textual gradients from "picking" incorrect model outputs, thereby mitigating the noise introduced by correctly handled examples. Second, the regularized verification step treats each resulting prompt update as a candidate, which is "picked" only if it passes a preservation test on a fixed holdout set of general examples, ensuring that targeted improvements do not compromise broad robustness. Experiments on several complex instruction-following and reasoning benchmarks demonstrate that our framework drastically reduces optimization instability, prevents performance degradation on general test cases, and consistently finds more robust prompts than standard iterative methods. Our work provides a principled approach to harnessing textual gradients with a high-quality learning signal and preventing specialization-induced degradation, thus enabling a more stable and effective methodology for automated prompt optimization.

## 1 INTRODUCTION

Large Language Models (LLMs) have demonstrated remarkable capabilities across a wide range of applications, from complex reasoning (Zhang et al., 2025; Yao et al., 2023; Kojima et al., 2022) to acting as the backbone for autonomous agents (Wang et al., 2024; Schick et al., 2023; Park et al., 2023). However, their performance is sensitive to the input prompt. Manually engineering prompts to elicit optimal performance is a tedious process that is often unpredictable and difficult to scale. While automated prompt optimization (APO) methods (Zhang et al., 2024a; Jain & Jindal, 2025) offer a promising alternative, they face challenges due to the stochasticity of the model's outputs. First, the optimization is undermined by the inherent non-determinism of LLM inference. Even with deterministic decoding (zero temperature), low-level computational variations in floating-point arithmetic and parallelization introduce stochasticity to the model's outputs (Whitehead & Fit-Florea, 2017). This variance creates a noisy evaluation landscape, making it difficult to reliably determine if a prompt update is genuinely effective. Second, this issue is amplified in methods that use a chain of LLM calls for feedback and updates, such as textual gradients (Yuksekgonul et al., 2024). A minor variance in the initial output can be magnified as it passes through the evaluator and optimizer LLMs, a form of cascading variance (Dohan et al., 2022).

To explore this process, early approaches explored the vast prompt space using search algorithms, including Monte Carlo Tree Search (e.g., PromptAgent (Wang et al., 2023)), genetic algorithms (e.g., GPS (Xu et al., 2022), EvoPrompt (Guo et al., 2023)), and discrete editing methods (e.g., GRIPS (Prasad et al., 2022), COPLE (Zhan et al., 2024)). While innovative, these methods often struggle with the semantic complexity of language and can be sample-inefficient. More recently, iterative refinement using *textual gradients* has emerged as the state-of-the-art (Zhang et al., 2024b; Pryzant et al., 2023; Yuksekgonul et al., 2024; Yu et al., 2025).

However, in this work, we identify and analyze two fundamental sources of instability inherent in this paradigm. First, we find that the quality of the textual gradient is highly dependent on the correctness of the initial output. As shown in Figure 1 (a), generating feedback from examples that the model already handles correctly produces a low-signal and high-noise gradient, often leading to destructive edits. While this observation suggests that the solution is to optimize exclusively on the high-signal feedback from failed cases, we find it is not this straightforward, because repeated refinement on these hard examples leads to over-specialization, sacrificing the prompt's general applicability on simpler tasks.

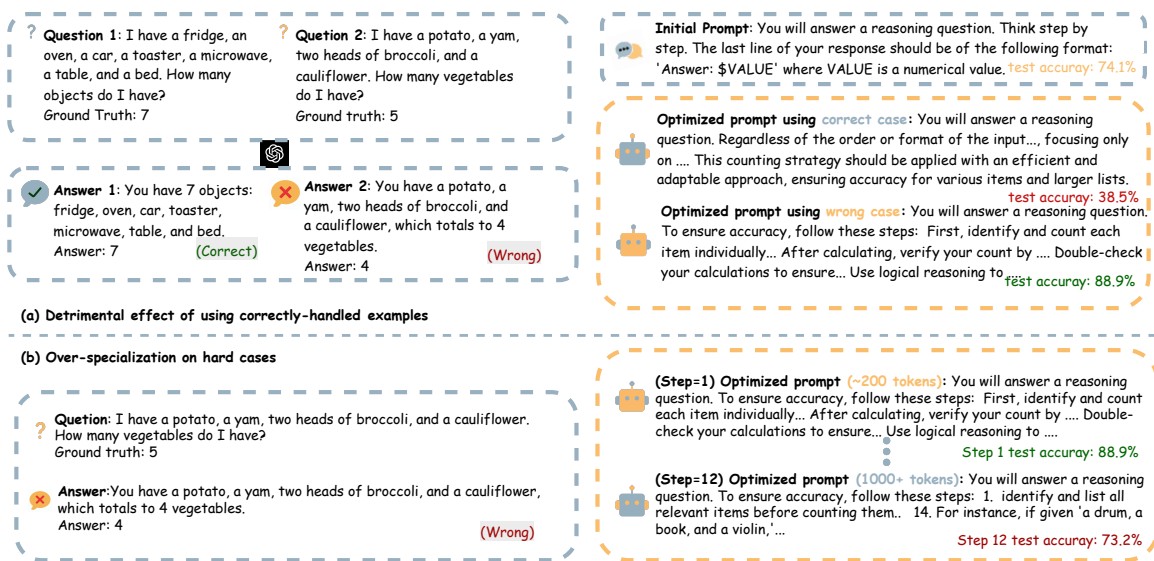

Figure 1: (a) The detrimental effect of using correctly-handled examples. Using a single prompt with a baseline accuracy of 74.1%, the executor LLM correctly answers Question 1 but fails on Question 2. Training new prompts through wrong cases can significantly improve the prompt accuracy (88.9%) compared to the correct one (38.5%). This is because correct cases produce a random gradient. Our work analyzes how the feedback derived from these different outcomes critically impacts the stability and effectiveness of the optimization process. (b) The over-specialization example on hard cases. It repeatedly refines the prompt on both failed and successful cases without verification, leading to overfitting. While an initial optimization step boosts accuracy to 88.9%, continued refinement over 12 steps results in an over-specialized prompt (1000+ tokens) with significantly degraded final accuracy (73.2%).

Second, we find that optimizing exclusively on difficult cases is surprisingly unstable. While initial gradients are corrective, performance degrades sharply after just a few iterations, as shown in Figure 1 (b). This occurs because the optimizer attempts to solve hard examples by adding more specific constraints and multi-step procedures into the prompt. For instance, to solve a complex object-counting problem, the prompt might be amended with explicit rules like, "First, list every potential object. Second, categorize each object. Third, create a final count based only on valid categories." While this rigid algorithm is effective for the targeted hard case, it becomes overly redundant for a simple case like "count the number of apples." For simpler inputs, the verbose and complex instructions can confuse the executor model or lead to inefficient reasoning paths, thereby degrading its performance. Consequently, the prompt becomes over-specialized at the cost of its general capabilities.

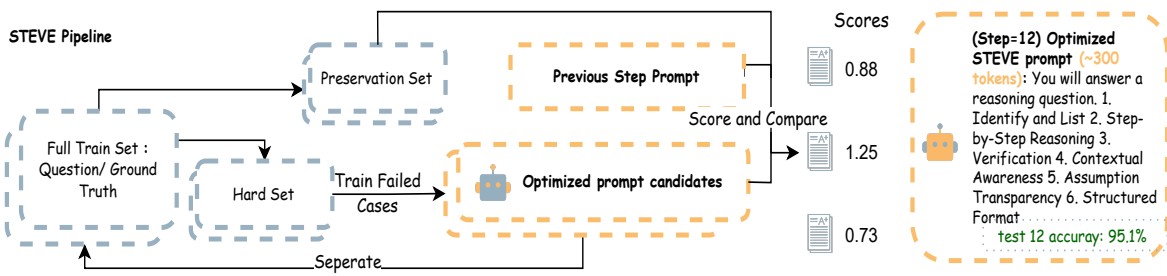

Figure 2: Our proposed STEVE pipeline's Error-driven Refinement mechanism can optimize a prompt based on failure cases by iteratively separating the training data into a Hard Set and a Preservation Set. At each step, Regularized Verification generates and scores multiple candidates using a regularized objective and select the best. This process avoids overfitting, producing a concise (~300 tokens) and highly effective final prompt that achieves 95.1% accuracy.

To address these two instabilities, we propose **STEVE: Stabilizing Textual Gradients via Error-Driven Refinement and Regularized Verification**. Our framework, as shown in Figure 2, introduces two core mechanisms. First, our **Error-Driven Refinement** strategy ensures a high-quality learning signal by exclusively generating textual gradients from model failures, filtering out the noise from correct examples. Second, to counter over-specialization, our **Regularized Verification** mechanism acts as a gate. It validates each candidate prompt against a holdout set of general examples, accepting an update only if the gain on the complex case does not compromise overall robustness. Together, these components create a stable optimization loop that effectively balances specialization and generalization.

Our contributions are threefold:

- We are the first to systematically identify and analyze two primary sources of instability in iterative prompt optimization: the generation of noisy, destructive gradients from correctly-handled examples, and the rapid overfitting that occurs when optimizing exclusively on model failures.
- We propose **STEVE**, a simple yet effective framework that directly counteracts these instabilities in iterative prompt optimization, comprising two core mechanisms. Error-Driven Refinement ensures a high-quality learning signal by generating gradients only from model failures, while Regularized Verification acts as a novel gating mechanism that accepts a prompt update only if it preserves performance on a general holdout set, explicitly preventing over-specialization on hard cases after multiple iterations.
- We demonstrate through extensive experiments on complex reasoning benchmarks that our framework leads to a more stable and effective optimization process. STEVE consistently discovers robust and concise prompts that achieve state-of-the-art performance, preventing the random texual gradient that plagues standard iterative prompt optimization methods.

## 2 RELATED WORK

**Automated Prompt Optimization** Manual prompt optimization methods (Wei et al., 2022) reveal that fine-tuning prompts can further improve the performance of LLMs and reduce token cost. They offer a baseline for automated prompt optimization methods, but fail to apply to extensive areas. The majority of automated prompt optimization work can be divided into two parts. Soft prompt tuning-based methods (Lester et al., 2021; Hu et al., 2022; Liu et al., 2024) utilize task-specific latent embeddings, limited by closed-source black-box LLMs. Discrete token search approaches further alleviate the availability by using textual (Wang et al., 2023; Do et al., 2024; Sinha et al., 2024) or numerical (Zhou et al., 2022; 2023b; Deng et al., 2022; Zhang et al., 2024a) signal as feedback. The different types of evaluation feedback provide ways to identify promising prompt candidates.

**Textual gradient-based learning** Taking advantage of textual gradient to guide the process of prompt optimization, ProTeGi (Pryzant et al., 2023) and TextGrad (Yuksekgonul et al., 2024) emerge as the state-of-art method in the domain of prompt optimization. These methods simulate a gradient in the discrete space of text, allowing for a more directed and efficient search for better prompts. The core idea is to use a separate LLM as an evaluator. When a target LLM produces an incorrect or suboptimal output based on the current prompt, the critic LLM is prompted to provide a natural language critique. This critique, which explains the flaws in the output and suggests improvements, is treated as a textual gradient. It provides a semantic direction for how the prompt should be edited. REVOLVE(Zhang et al., 2024b) further refines this by not only considering the immediate textual gradient but also the historical evolution of responses. This allows for a more nuanced optimization that is analogous to second-order methods, leading to more robust and efficient prompt improvements.

**Quality of Learning Signals** A core challenge in iterative optimization is the quality of the guiding signal. Our Error-Driven Refinement mechanism is motivated by established principles in active learning and information theory(Nguyen et al., 2021; Li et al., 2024). The central tenet of active learning (Settles, 2009) is that a model learns most efficiently from examples it finds difficult or uncertain about. Correctly handled examples provide a low-information signal. Forcing an LLM to generate feedback on a correct output can lead to random or stylistic critiques that act as noise, degrading the prompt rather than improving it. This aligns with findings from instruction tuning, LIMA(Zhou et al., 2023a) demonstrates that a small set of high-quality, diverse data is far more effective than a large volume of noisy or low-quality data.

## 3 METHOD

TextGrad (Yuksekgonul et al., 2024) is a prominent iterative method that leverages textual gradients for prompt optimization. While we use TextGrad as our foundational optimizer, the framework we introduce is primarily data-driven and largely agnostic to the specific gradient generation process, making it broadly applicable to any iterative and tex-

tual gradient-based method (Pryzant et al., 2023; Yang et al., 2023; Zhang et al., 2024b). The central contribution of our work is a novel framework designed to stabilize this optimization process by directly addressing two critical failure modes we identified: the generation of noisy, often destructive gradients from correctly handled examples, and over-specialization on hard cases. In the following sections, we will detail the two core components of our solution: Error-Driven Refinement and Regularized Verification.

## 3.1 PRELIMINARIES: ITERATIVE OPTIMIZATION WITH TEXTUAL GRADIENTS

The goal of prompt optimization is to find an optimal instruction, or prompt $p^*$, that maximizes the performance of an LLM $M$ across a given task distribution $\mathcal{D}$. Formally, this can be expressed as following:

$$p^* = \arg \max_{p \in p_{\text{space}}} \mathbb{E}_{(x,y) \sim \mathcal{D}}[\mathcal{S}(M(p,x), y)] \tag{1}$$

where $x$ is a question, $y$ is the ground-truth answer, and $\mathcal{S}$ is a task-specific evaluation metric.

To navigate this challenge, recent works Zhang et al. (2024b); Pryzant et al. (2023); Cui et al. (2024) have proposed using an iterative refinement process guided by **textual gradients**. This approach typically employs a multi-agent framework. At each iteration $t$, the process unfolds as follows:

(a) **Forward Pass Generation:** A *executor LLM* uses the current prompt, $p_t$, to process an input, $x$, and generate an output, $y = M_{executor}(p_t, x)$.

(b) **Feedback (Gradient Calculation):** A powerful *evaluator LLM* evaluates the output $y'$ and calculate the loss from $y$ and ground truth $y_{truth}$. The evaluator generates a "textual gradient," $g_{\text{text}}$. This gradient is a natural language critique that explains the failure and provides actionable advice for improving the prompt, $g_{next} = M_{evaluator}(p_t, x \mid y, y_{truth})$

(c) **Update:** An *optimizer LLM*, conditioned on the original prompt $p_t$ and the textual gradient $g_{\text{text}}$, synthesizes an improved prompt, $p_{t+1}$. This update step can be represented as: $p_{t+1} = M_{optimizer}(p_t, g_{\text{text}})$.

This iterative loop allows for semantic and non-differentiable improvements to the prompt. While this powerful paradigm forms the backbone of our method, we show that this standard approach suffers from significant instability, which often prevents it from converging to a robust and high-performing solution. The following sections will analyze and address these instabilities.

## 3.2 THE INSTABILITY OF TEXTUAL GRADIENT OPTIMIZATION

While the iterative process described in Section 3.1 provides a powerful framework, its practical application is hindered by gradient instability. Based on our experiment, we characterize two primary sources of this instability, which motivate our proposed control mechanisms.

**Noisy Gradients from Correct Examples.** Our initial investigation reveals a stark contrast in the utility of feedback based on the correctness of an output. As illustrated in Figure 3, a single round of optimization using feedback from previously *failed* cases yields a significant performance improvement. Conversely, using feedback from *correctly-handled* cases results in a sharp drop in accuracy. This observation leads us to hypothesize that textual gradients generated from correct examples are noisy and often counter-productive. We validate this hypothesis in our ablation study (Table 2), where a model trained exclusively on correct examples not only fails to improve but even sees its performance degrade below the initial baseline. This suggests that the evaluator LLM struggles to extract a meaningful, generalizable improvement signal from successful outputs, instead producing feedback with noisy gradients that acts as a random perturbation.

**Over-Specialization on Hard Cases.** Simply optimizing on failed cases, however, is not a complete solution. We observe that sustained, multi-round optimization exclusively on hard examples is also unstable, with performance degrading after an initial phase of improvement. This decay is a symptom of overfitting in the prompt space. As we show with a qualitative example in the Appendix A.3, the prompt becomes progressively more verbose and convoluted over iterations, accumulating an excessive number of specific constraints and detailed steps. While these highly specific instructions may resolve the targeted hard cases, they make the prompt brittle and less effective for simpler, more general problems. The added complexity is detrimental to its overall generalization, motivating a mechanism to explicitly control this trade-off.

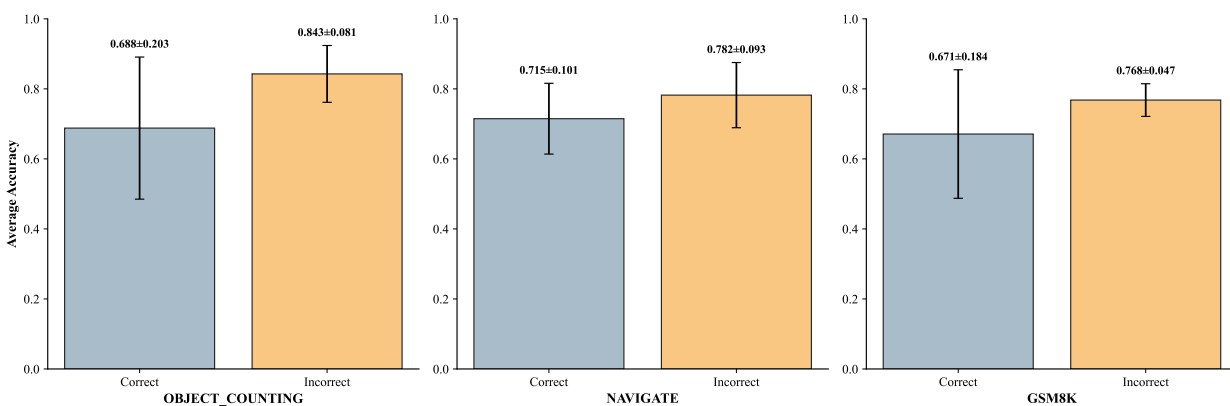

Figure 3: A comparison of single-step prompt refinement on three reasoning benchmarks Object Counting, Navigate and GSM8k (Suzgun et al., 2022; Cobbe et al., 2021) . The bars show the final average accuracy of a prompt optimized for one round using feedback from either a batch of previously correct examples or a batch of previously incorrect examples. Across all datasets, optimizing on incorrect cases consistently yields a prompt with significantly higher accuracy, demonstrating that failures provide a superior learning signal.

### 3.3    COMPONENT 1: ERROR-DRIVEN REFINEMENT FOR HIGH-QUALITY GRADIENTS

To address the instability caused by noisy gradients, we introduce our first mechanism: **Error-Driven Refinement**. Motivated by the idea of Prioritized Experience Replay(Schaul et al., 2015; Ma et al., 2022) in the field of Reinforcement Learning, the model can focus more on transitions where its value prediction was wrong, effectively balancing the training process by focusing on hard or more informative samples. This principle that focuses on hard examples is critical for effective learning and is well-established in the machine learning literature. Seminal examples include boosting algorithms like AdaBoost(Freund & Schapire, 1997), which iteratively re-weights misclassified data points, and online hard example mining (OHEM)(Shrivastava et al., 2016) in object detection, which explicitly trains on the most challenging examples. The core principle of this strategy is to ensure a high signal-to-noise ratio in the learning process by exclusively generating textual gradients from examples that the executor model fails to handle correctly.

We formalize this by partitioning the training set, $\mathcal{D}_{\text{train}}$, based on the performance of the current prompt, $p_t$. Specifically, we define a "hard case pool," $\mathcal{D}_{\text{hard}}$, as the subset of training examples where the executor model's output is deemed incorrect by the scoring function:

$$\mathcal{D}_{\text{hard}}(p_t) = \{(x, y) \in \mathcal{D}_{\text{train}} \mid \mathcal{S}(M_{\text{executor}}(p_t, x), y) < \tau\} \tag{2}$$

where $\tau$ is a predefined success threshold (typically 1 for exact match tasks).

At each optimization iteration $t$, instead of sampling from the entire training distribution, our method samples an instance $(x_i, y_i)$ exclusively from this dynamically defined hard case pool, $\mathcal{D}_{\text{hard}}(p_t)$. A textual gradient, $g_{\text{text}}$, is then generated based on the model's failure on this specific instance.

By design, this error-driven strategy acts as a powerful information filter. It guarantees that every textual gradient used for an update is a high-signal, corrective piece of feedback derived from a clear failure. This eliminates the random walk behavior caused by the low-signal, high-noise gradients generated from correct examples, thereby solving the first source of instability and providing a solid foundation for targeted prompt improvement.

### 3.4    COMPONENT 2: REGULARIZED VERIFICATION FOR PRESERVING GENERALIZATION

While Error-Driven Refinement ensures that each textual gradient is informative, it does not prevent the optimization from over-specializing. To address this second instability, we introduce our second component: a **Regularized Verification** mechanism. This mechanism functions as a gate, evaluating each proposed prompt update to ensure that improvements in specialization do not come at an unacceptable cost to generalization.

Our approach is conceptually inspired by regularization-based methods in continual learning designed to combat catastrophic forgetting, most notably Elastic Weight Consolidation (EWC) (Kirkpatrick et al., 2017). EWC adds a quadratic penalty to the loss function to discourage modifications to network weights that are critical for performance on previously learned tasks. Analogously, our Regularized Verification mechanism treats the performance degradation on the

Generalization Preservation Set as a direct penalty against forgetting. However, unlike the continuous, differentiable parameter space of neural networks where penalties can be directly integrated into a loss function, the prompt space is discrete and symbolic. Therefore, instead of modifying a loss function, we implement this regularization principle as a discrete verification step: a "generate-and-select" mechanism where candidate prompts are explicitly evaluated for their trade-off between specialization and generalization.

Central to this process is the **Generalization Preservation Set**, $\mathcal{D}_{\text{preserve}}$, a fixed holdout set of representative examples sampled from the training distribution. This set is typically constructed from examples that the current prompt already handles correctly, thus representing the current baseline capabilities we aim to preserve.

$$\mathcal{D}_{\text{preserve}} \subset \{(x, y) \in \mathcal{D}_{\text{train}} \mid \mathcal{S}(M_{\text{executor}}(p_0, x), y) \geq \tau\} \tag{3}$$

The verification process is integrated into a multi-stage refinement loop. First, a batch of hard cases $\{d_h\}_1^b \subset \mathcal{D}_{\text{hard}}$ is sampled. An evaluator model synthesizes a textual gradient, $g_{\text{batch}}$, that summarizes the common failure modes across this batch. Using this gradient, an updater model generates a set of $n$ diverse candidate prompts, $\{p_{\text{cand},i}\}_1^n$. The best candidate is then chosen based on our regularized objective. The formal update rule is:

$$p_{t+1} = \begin{cases} p_{\text{best\_cand}} & \text{if Improvement}(p_{\text{cand}}, d_{\text{hard}}) > \lambda \cdot \text{Regression}(p_{\text{cand}}, \mathcal{D}_{\text{preserve}}) \\ p_t & \text{otherwise} \end{cases} \tag{4}$$

where $\lambda \geq 0$ is the regularization hyperparameter that controls the trade-off. $\lambda$ is annealed per step so that it can balance the generalization and specialization. By including $p_t$ in the set of choices, we ensure an update only occurs if a new candidate offers a better trade-off than the status quo. The full process is detailed in Algorithm 1.

---

**Algorithm 1** The STEVE Algorithm (Stabilizing Textual Gradients via Error-Driven Refinement and Regularized Verification)

---

**Require:** Initial prompt $p_0$, training data $\mathcal{D}_{\text{train}}$, iterations $T$, regularization $\lambda$.
**Require:** Preservation sample size $k$, hard-case batch size $b$, num candidates $n$.
**Ensure:** Optimized prompt $p_T$.
1: Initialize $\mathcal{D}_{\text{preserve}} \subset \{(x, y) \in \mathcal{D}_{\text{train}} \mid \mathcal{S}(M(p_0, x), y) \geq \tau\}$.
2: $p_t \leftarrow p_0$
3: **for** $t = 0$ to $T - 1$ **do**
4:     Define hard case pool $\mathcal{D}_{\text{hard}}(p_t) = \{(x, y) \in \mathcal{D}_{\text{train}} \mid \mathcal{S}(M(p_t, x), y) < \tau\}$.
5:     **if** $|\mathcal{D}_{\text{hard}}(p_t)| < b$ **then**
6:         **break**                                            ▷ Not enough errors to fix.
7:     **end if**
8:     **for** $j = 1$ to $n$ **do**
9:         Sample a batch $D_{\text{batch}} = \{(x_i, y_i)\}_{i=1}^b$ from $\mathcal{D}_{\text{hard}}(p_t)$.
10:        Forward pass $\{y_i'\}_{i=1}^b \leftarrow \{M_{\text{executor}}(p_t, x_i)\}_{i=1}^b$
11:                                   ▷ *Error-Driven Refinement (Section 3.3)*
12:        Synthesize a batch gradient $g_{\text{batch}} = M_{\text{evaluator}}(p_t, D_{\text{batch}}, \{y_i'\}_{i=1}^b)$.
13:        Generate candidate prompt $p_{\text{cand}} = M_{\text{optimizer}}(p_t, g_{\text{batch}})$.
14:                                   ▷ *Regularized Verification (Section 3.4)*
15:        Sample $D_{\text{sample}} = \{(x_j, y_j)\}_{j=1}^k$ from $\mathcal{D}_{\text{preserve}}$.
16:        best_score $\leftarrow 0$
17:                                 ▷ Objective score of the current prompt is 0.
18:        Improvement $\leftarrow \frac{1}{b} \sum_{i=1}^b (\mathcal{S}(M_{\text{executor}}(p_{\text{cand}}, x_i), y_i) - \mathcal{S}(M_{\text{executor}}(p_t, x_i), y_i))$.
19:        Regression $\leftarrow \frac{1}{k} \sum_{j=1}^k (\mathcal{S}(M_{\text{executor}}(p_t, x_j), y_j) - \mathcal{S}(M_{\text{executor}}(p_{\text{cand}}, x_j), y_j))$.
20:        objective_score $\leftarrow$ Improvement $- \lambda \cdot \max(0, \text{Regression})$.
21:        **if** objective_score $>$ best_score **then**
22:            best_score $\leftarrow$ objective_score
23:            $p_{t+1} \leftarrow p_{\text{cand}}$
24:        **end if**
25:     **end for**
26: **end for**
27: **return** $p^* = \arg\max_{p \in \{p_0, \dots, p_T\}} \mathbb{E}_{(x,y) \sim \mathcal{D}_{\text{test}}}[\mathcal{S}(M_{\text{executor}}(p, x), y)]$.

---

## 4 EXPERIMENT

We conduct a comprehensive evaluation to validate our proposed framework, STEVE. Our experiments are designed to: (1) assess whether STEVE significantly outperforms established baselines and state-of-the-art prompt optimization methods across diverse reasoning tasks; (2) quantify the contribution of our core components via ablation studies; and (3) analyze the impact of evaluator model quality on the final optimized prompt. More results and settings are available in the section Appendix. All the source code and datasets will be made available to the public.

### 4.1 DATASETS AND TASKS

To ensure a thorough assessment of our method's generalization capabilities, we evaluate it on 10 challenging benchmarks spanning four distinct reasoning domains. For mathematical reasoning, we use **GSM8k** (Cobbe et al., 2021) and **MultiArith** (Roy & Roth, 2016), which test multi-step numerical problem-solving. For complex commonsense reasoning, we assess multi-hop logical deduction using **StrategyQA** (Geva et al., 2021) and the **Navigate** task from Big-Bench Hard (BBH) (Suzgun et al., 2022). To evaluate precise procedural execution in symbolic and procedural reasoning, we use four tasks from BBH: **Object Counting**, **Penguins in a Table**, **Geometric Shapes**, and **Date Understanding**. Finally, to assess expert-level knowledge reasoning on domain-specific topics, we use the **College Physics** and **Machine Learning** subsets from the Massive Multitask Language Understanding (MMLU) benchmark (Hendrycks et al., 2020).

### 4.2 EXPERIMENTAL SETUP

**Baselines.** We compare our method, STEVE, against four baselines. **Zero-shot CoT** (Wei et al., 2022) serves as the standard initial prompt ($P_0$) for all iterative methods. **Three-shot CoT** represents a strong, manually crafted few-shot baseline. While these static baselines keep the prompt unchanged, we also compare against two state-of-the-art iterative optimization methods. The first is **TextGrad** (Yuksekgonul et al., 2024), the textual gradient method our work builds upon, optimize and refine the outputs of LLM output by using textual feedback as a gradient. The second is **REVOLVE** (Zhang et al., 2024b), an advanced method that enriches the learning signal by analyzing the historical evolution of responses to a given problem over time, creating a more nuanced update process analogous to a second-order optimization method.

**Implementation Details.** For a fair comparison, all iterative methods (TextGrad, REVOLVE, and STEVE) use the same setup. The executor model is `gpt-3.5-turbo-0125` and the default evaluator/optimizer models are `gpt-4o`, `gemini-2.5-flash`, and `gpt-5`. Optimization is run for $T = 12$ steps with a hard-case batch size of $b = 4$. The decoding temperature is set to 0.0 for all models to ensure reproducibility. For our method, the regularization parameter $\lambda$ is initialized at 1.5 and is annealed by +0.1 at each step. The primary evaluation metric is accuracy, and we report the average over three independent optimization runs.

### 4.3 RESULTS

We present our primary findings in Table 1, which compares the final accuracy of STEVE against all baselines across our 10 benchmark datasets and three evaluator/optimizer models. The results demonstrate that STEVE consistently and significantly outperforms both static baselines (Zero-shot and Three-shot CoT) and state-of-the-art iterative optimization methods on the vast majority of configurations. Averaged across all 30 settings, STEVE achieves an absolute improvement of over 15% compared to the initial Zero-shot CoT prompt and outperforms the strongest iterative baseline, REVOLVE, by an average of 3.5%.

The significant performance gap between STEVE and TextGrad, which also uses textual gradients, highlights the importance of our stability mechanisms. While both methods start from the same initial prompt, TextGrad's optimization trajectory can be volatile. This is particularly evident on tasks like **Geometric Shapes** with the `gpt-5` executor, where TextGrad's accuracy collapses from 41.1% to 33.9%, indicating a destructive update sequence. In stark contrast, STEVE's combination of **Error-Driven Refinement** (filtering for high-quality signals) and **Regularized Verification** (preventing generalization loss) navigates the optimization landscape more effectively, achieving a stable improvement to 44.2%. This demonstrates our framework's ability to prevent overfitting and harness textual gradients reliably.

STEVE shows particularly strong performance on tasks requiring complex procedural or symbolic reasoning. For instance, on **Navigate** and **Penguins in a Table** with the `gemini-2.5-flash` evaluator/optimizer, STEVE achieves gains of over 27% and 30% respectively. We hypothesize that these tasks involve discovering non-obvious, robust strategies that are easily missed by unstable optimizers. STEVE's verification mechanism allows it to safely explore

Table 1: Main results on **Mathematical, Commonsense, Symbolic/Procedural,** and **Expert-Level Knowledge Reasoning** benchmarks, broken down by executor model. All iterative methods are optimized for 12 steps. We report accuracy with the absolute improvement (↑↓) over the Zero-shot CoT baseline. The best performance for each dataset is highlighted in **bold**. Underlined indicates second-best performance.

| Category | Task | Executor Model | Baselines | | Iterative Optimization Methods | | |
|---|---|---|---|---|---|---|---|
| | | | Zero-shot | Three-shot | TextGrad | REVOLVE | STEVE (Ours) |
| Math | GSM8k (Cobbe et al., 2021) | gpt-4o | 76.3 | 73.1 | 82.5 ↑+6.2 | 82.8 ↑+6.5 | **86.2** ↑+9.9 |
| | | gemini-2.5-flash | 73.8 | 74.2 | 81.1 ↑+7.3 | **83.9** ↑+10.1 | 82.9 ↑+9.1 |
| | | gpt-5 | 73.2 | 77.5 | 78.6 ↑+5.4 | 81.7 ↑+8.5 | **84.3** ↑+11.1 |
| | MultiArith (Roy & Roth, 2016) | gpt-4o | 84.5 | 84.1 | 88.6 ↑+4.1 | 98.2 ↑+13.7 | **98.9** ↑+14.4 |
| | | gemini-2.5-flash | 82.9 | 84.3 | 98.1 ↑+15.2 | 98.4 ↑+15.5 | **98.7** ↑+15.8 |
| | | gpt-5 | 84.0 | 84.6 | **100.0** ↑+16.0 | **100.0** ↑+16.0 | **100.0** ↑+16.0 |
| Commonsense | StrategyQA (Geva et al., 2021) | gpt-4o | 88.7 | 91.3 | 90.1 ↑+1.4 | 90.5 ↑+1.8 | **93.2** ↑+4.5 |
| | | gemini-2.5-flash | 85.1 | 90.6 | 88.8 ↑+3.7 | 89.4 ↑+4.3 | **91.9** ↑+6.8 |
| | | gpt-5 | 88.2 | 94.5 | 91.7 ↑+3.5 | 93.3 ↑+5.1 | **95.8** ↑+7.6 |
| | Navigate (Suzgun et al., 2022) | gpt-4o | 60.1 | 77.8 | 88.2 ↑+28.1 | 94.1 ↑+34.0 | **95.6** ↑+35.5 |
| | | gemini-2.5-flash | 68.7 | 83.6 | 66.9 ↓-1.8 | 90.4 ↑+21.7 | **96.2** ↑+27.5 |
| | | gpt-5 | 62.7 | 83.1 | 78.6 ↑+15.9 | **86.1** ↑+23.4 | 83.9 ↑+21.2 |
| Symbolic | Object Counting (Suzgun et al., 2022) | gpt-4o | 77.9 | 82.2 | 87.1 ↑+9.2 | 90.3 ↑+12.4 | **95.7** ↑+17.8 |
| | | gemini-2.5-flash | 75.4 | 81.8 | 72.5 ↓-2.9 | 81.1 ↑+5.7 | **83.6** ↑+8.2 |
| | | gpt-5 | 79.0 | 87.5 | 80.6 ↑+1.6 | 82.4 ↑+3.4 | **91.2** ↑+12.2 |
| | Penguins in a Table (Suzgun et al., 2022) | gpt-4o | 80.8 | 83.3 | 93.1 ↑+12.3 | 96.0 ↑+15.2 | **96.5** ↑+15.7 |
| | | gemini-2.5-flash | 66.2 | 90.7 | 90.9 ↑+24.7 | 90.2 ↑+24.0 | **96.4** ↑+30.2 |
| | | gpt-5 | 63.5 | 90.1 | 93.6 ↑+30.1 | 86.8 ↑+23.3 | **96.7** ↑+33.2 |
| | Geometric Shapes (Suzgun et al., 2022) | gpt-4o | 39.4 | 31.8 | 36.7 ↓-2.7 | 55.2 ↑+15.8 | **62.9** ↑+23.5 |
| | | gemini-2.5-flash | 36.6 | 36.1 | 48.3 ↑+11.7 | 65.5 ↑+28.9 | **66.0** ↑+29.4 |
| | | gpt-5 | 41.1 | 32.7 | 33.9 ↓-7.2 | 42.4 ↑+1.3 | **44.2** ↑+3.1 |
| | Date Understanding (Suzgun et al., 2022) | gpt-4o | 67.3 | 73.9 | 75.1 ↑+7.8 | 76.2 ↑+8.9 | **76.6** ↑+9.3 |
| | | gemini-2.5-flash | 70.8 | 68.2 | 74.4 ↑+3.6 | 74.9 ↑+4.1 | **80.5** ↑+9.7 |
| | | gpt-5 | 70.1 | 72.7 | 74.3 ↑+4.2 | 77.8 ↑+7.7 | **84.0** ↑+13.9 |
| Expert | College Physics (Hendrycks et al., 2020) | gpt-4o | 57.6 | 52.1 | 61.4 ↑+3.8 | 66.9 ↑+9.3 | **71.2** ↑+13.6 |
| | | gemini-2.5-flash | 61.0 | 52.8 | 57.3 ↓-3.7 | 61.5 ↑+0.5 | **66.7** ↑+5.7 |
| | | gpt-5 | 52.3 | 57.9 | 57.5 ↑+5.2 | 61.8 ↑+9.5 | **66.1** ↑+13.8 |
| | Machine Learning (Hendrycks et al., 2020) | gpt-4o | 34.2 | 39.8 | 60.1 ↑+25.9 | 60.3 ↑+26.2 | **60.7** ↑+26.5 |
| | | gemini-2.5-flash | 43.5 | 34.6 | 52.9 ↑+9.4 | **60.4** ↑+16.9 | 56.2 ↑+12.7 |
| | | gpt-5 | 47.7 | 39.1 | 56.6 ↑+8.9 | 43.3 ↓-4.4 | **60.9** ↑+13.2 |

and lock in complex heuristics that generalize well, whereas other methods may discard them or overfit to a brittle solution. This finding further demonstrates the robustness and effectiveness of our STEVE method.

While broadly successful, the margin of improvement varies. On tasks such as **MultiArith**, where the initial prompt is already quite effective, the gains are more modest as there is less room for optimization. Notably, on the **Machine Learning** MMLU task with gemini-2.5-flash, REVOLVE slightly outperforms STEVE. We conjecture that for certain highly knowledge-intensive domains, REVOLVE's approach of tracking response evolution may be more effective at eliciting factual recall. It is also possible that STEVE's conservative verification gate prevented a risky but ultimately beneficial update for this specific configuration. This highlights a potential area for future work in dynamically adjusting the regularization strength based on task type.

## 4.4 ABLATION STUDIES

To isolate and validate the contributions of our framework's core components, we conduct a series of ablation studies on a representative subset of datasets: GSM8k, StrategyQA, and Object Counting. The results, summarized in Table 2, confirm our design choices.

Table 2: Ablation studies on the core components of STEVE. We report performance on a representative subset of datasets. The results demonstrate the importance of both the regularized verification gate and the error-driven learning signal. The best performance for each dataset is highlighted in **bold**.

| Model Variant | GSM8k | StrategyQA | Object Counting |
|---|---|---|---|
| Initial CoT Prompt | 76.3 | 88.7 | 77.9 |
| *Analysis of Regularized Verification* | | | |
| w/o Verification | 84.1 | 91.5 | 92.2 |
| *Analysis of the Error-Driven Refinement* | | | |
| w/ Full Dataset | 82.5 | 90.1 | 87.1 |
| w/ Correct-Only | 76.5 | 86.8 | 76.1 |
| STEVE (Full Model) | **86.2** | **93.2** | **95.7** |

**Contribution of Regularized Verification.**     First, to demonstrate the necessity of our verification gate, we evaluate **STEVE w/o Verification**. In this variant, we remove the Verification module, meaning the candidate prompts are trained by the hard-case batch without a regression check. This variant underperforms the full STEVE model, confirming that without explicit regularization to safeguard general capabilities, the prompt quickly overfits to the hard cases it is trained on.

**Analysis of the Error-Driven Signal.**     Second, we validate our core hypothesis that the learning signal must be error-driven. We test two variants: (1) **STEVE w/ Full Dataset**, which trains on the complete, unfiltered training set containing both correct and incorrect examples, and (2) **STEVE w/ Correct-Only**, a control experiment that trains exclusively on examples the model already handles correctly. The first variant suffers from the noisy gradients of correct examples and performs poorly. The second variant consistently degrades the prompt's performance, often below the initial baseline. These results provide strong evidence that a high-quality, corrective signal derived exclusively from errors is essential for stable and effective optimization.

## 5 CONCLUSION

In this work, we address the critical instability of textual gradient-based prompt optimization by reframing it as a dual-selection problem we call **"Pick Your Textual Gradients."** To solve the failure modes of noisy feedback and over-specialization, we introduced STEVE, a framework that carefully picks its updates at two key stages. First, our **Error-Driven Refinement** mechanism ensures a high-quality learning signal by selectively score feedback generated exclusively from previously failed cases, filtering out the noise from correct examples. Second, our **Regularized Verification** step addresses overfitting by selecting the best candidate prompt—not just the one that solves the hard case, but the one with the best generalization performance as measured by a regularized objective. Our experiments demonstrated that this principled approach of carefully picking both the learning signal and the final update allows STEVE to consistently discover more robust and effective prompts than standard methods, paving the way for more reliable automated prompt engineering.

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

# A APPENDIX

## A.1 EXPERIMENTAL SETUP DETAILS

All the source code and datasets will be made available to the public. This section provides additional details regarding our experimental setup to ensure full reproducibility.

### A.1.1 DATASET DETAILS

Table 3 provides a summary of the datasets used in our evaluation. The column "Train Set" refers to the number of examples available for the iterative optimization process (from which $\mathcal{D}_{hard}$ is sampled). The "Preservation Set" is a fixed subset of the training data used for our regularized verification, and the "Test Set" is used for final evaluation.

Table 3: Summary of dataset statistics used in our experiments.

| Dataset | Train Set Size | Test Set Size | Preservation Sample Set Size | Candidate Size |
|---|---|---|---|---|
| *Mathematical Reasoning* | | | | |
| GSM8k | 50 | 100 | 20 | 3 |
| MultiArith | 50 | 50 | 20 | 3 |
| *Complex Commonsense Reasoning* | | | | |
| StrategyQA | 50 | 100 | 20 | 3 |
| Navigate | 50 | 100 | 20 | 3 |
| *Symbolic & Procedural Reasoning* | | | | |
| Object Counting | 50 | 100 | 20 | 3 |
| Penguins in a Table | 87 | 30 | 20 | 3 |
| Geometric Shapes | 50 | 100 | 20 | 3 |
| Date Understanding | 50 | 100 | 20 | 3 |
| *Expert-Level Knowledge Reasoning* | | | | |
| College Physics | 50 | 100 | 20 | 3 |
| Machine Learning | 67 | 23 | 20 | 3 |

### A.1.2 MODEL AND API DETAILS

All experiments were conducted using API access to the respective language models. The specific model versions used are as follows:

- **Executor Model:** `gpt-3.5-turbo-0125`
- **Evaluator/Optimizer Models:** `gpt-4o` (version `gpt-4o-2024-05-13`), `gpt-5` (version `gpt-5-2025-08-07`), and `gemini-2.5-flash` (release data: June 17, 2025), whose exact version is not available .
- **Decoding Temerature:** 0.0
- **Top-p:** 0
- **Seed:** 42

### A.1.3 COMPUTATIONAL COST ANALYSIS

The primary computational cost of iterative prompt optimization methods like TextGrad and STEVE is the number of Large Language Model (LLM) API calls required. According to Table 4, we analyze this cost on the number of tokens and API calls..

Our method, STEVE, intentionally incurs a higher computational cost per iteration to ensure optimization stability. The additional cost arises from two main sources within our framework. First, instead of generating a single candidate, we generate $n$ diverse candidates to better explore the solution space. Second, and more significantly, our regularized verification step requires evaluating each of the $n$ candidates on a preservation set of size $k$. The dominant overhead of our method is therefore approximately $n \times (b + k)$ additional executor LLM calls per iteration compared to a non-verifying, single-candidate approach.

Table 4: Estimated token consumption and cost for a single STEVE optimization run on BBH Object Counting. Assumes $n = 3$ candidates and a preservation set sample size of $k = 20$. Costs are based on September 2025 pricing for `gpt-4o` (evaluator/optimizer) and `gpt-3.5-turbo-0125` (executor).

| Task | Total API Calls (Executor/Evaluator/Optimizer) | Est. Tokens (Executor/Evaluator/Optimizer) | Est. Total Cost (USD) |
|---|---|---|---|
| BBH Object Counting (1 Run) | 2298 / 60 / 20 | ~1,730,156 / ~36,260 / ~108,780 | ~$4.05 |

## A.2 IMPLEMENTATION DETAILS OF STEVE

### A.2.1 PROMPTS FOR EVALUATOR AND OPTIMIZER MODELS

Reproducibility of our method relies on the meta-prompts used to guide the evaluator and optimizer models. Below are the prompts used in our experiments.

---

**Evaluator Prompt**

"<OBJECTIVE_FUNCTION>Your goal is to give feedback and criticism to the variable given the above evaluation output."
"Our only goal is to improve the above metric, and nothing else. </OBJECTIVE_FUNCTION>"
"This conversation is part of a larger system. The <OUTPUT_OF_FUNCTION>was later used as {response_desc}."
"<OBJECTIVE_FUNCTION>Your goal is to give feedback to the variable to address the following feedback on the OUTPUT_OF_FUNCTION: {response_gradient} </OBJECTIVE_FUNCTION>"
"We are interested in giving feedback to the {variable_desc} "
"for this conversation. Specifically, give feedback to the following span "
"of text: <VARIABLE>"
"{variable_short} </VARIABLE>"
"Given the above history, describe how the {variable_desc} "
"could be improved to improve the <OBJECTIVE_FUNCTION>. Be very creative, critical, and intelligent."

---

**Optimizer Prompt**

"Here is the role of the variable you will improve: <ROLE>{variable_desc}</ROLE>."
"The variable is the text within the following span: <VARIABLE>{variable_short} </VARIABLE>"
"Here is the context and feedback we got for the variable:"
<CONTEXT>{variable_grad}</CONTEXT>
"Improve the variable ({variable_desc}) using the feedback provided in <FEEDBACK>tags."
"Send the improved variable "
"in the following format:"
{new_variable_start_tag}{{the improved variable}}{new_variable_end_tag}
"Send ONLY the improved variable between the <IMPROVABLE>tags, and nothing else."

---

### A.2.2 INITIAL PROMPTS ($P_0$)

All iterative optimization methods in our experiments began from a general Zero-shot Chain-of-Thought (CoT) prompt, $P_0$. This ensures that performance gains are a direct result of the optimization process. To accommodate specific output formats required by certain benchmarks, minor instructional text was added to a base prompt. Table 5 details the exact initial prompt used for each of the 10 datasets. No other task-specific modifications or in-context examples were included.

Table 5: Initial prompts ($P_0$) used for each benchmark in our experiments.

| Dataset | Initial Prompt ($P_0$) Text |
|---|---|
| *Mathematical Reasoning* | |
| GSM8k | `You will answer a reasoning question.  Think step by`
`step.`
`The last line of your response should be of the`
`following format:`
`'Answer:  $VALUE' where VALUE is a numerical value.` |
| MultiArith | `You will solve arithmetic word problems.  Think step`
`by step and output your final answer in the format`
`'Answer:  $NUMBER'.` |
| *Complex Commonsense Reasoning* | |
| StrategyQA | `Answer the following yes/no question.  Think step by step and`
`provide reasoning before answering.  The last line of your`
`response should be of the following format:  'Answer:  True'`
`or 'Answer:  False'.` |
| Navigate | `You will answer a reasoning question.  Think step by step.`
`The last line of your response should be of the following`
`format:  'Answer:  $VALUE' where VALUE is a numerical value.` |
| *Symbolic & Procedural Reasoning* | |
| Object Counting | `You will answer a reasoning question.  Think step by`
`step.`
`The last line of your response should be of the`
`following format:`
`'Answer:  $VALUE' where VALUE is a numerical value.` |
| Penguins in a Table | `You will answer a reasoning question.  Think step by`
`step.`
`The last line of your response should be of the`
`following format:`
`'Answer:  $VALUE' where VALUE is a numerical value.` |
| Geometric Shapes | `You will answer a reasoning question.  Think step by`
`step.`
`The last line of your response should be of the`
`following format:`
`'Answer:  $VALUE' where VALUE is a numerical value.` |
| Date Understanding | `Answer the following multiple choice question.  Think`
`step by step.`
`The last line must be 'Answer:  $LETTER'. LETTER must`
`be one of A, B, C, D, E, or F.` |
| *Expert-Level Knowledge Reasoning* | |
| College Physics | `You will answer multiple-choice questions.  Think step by`
`step.  The goal is to select the correct final answer from the`
`choices.` |
| Machine Learning | `You will answer multiple-choice questions.  Think step by`
`step.  The goal is to select the correct final answer from the`
`choices.` |

## A.3 A QUALILATIVE EXAMPLE OF OVER-SPECIALIZATION ON HARD CASES

Below is the final prompt optimized by hard cases without Regularized Verification on dataset StrategyQA. The prompt includes details about processes to solve specfic hard examples.

---

**Prompt on Hard Cases after 12 Steps Training**

"Answer the following yes/no question. Begin with a clear 'Answer: True' or 'Answer: False' statement. First, identify the core question and distinguish between primary and secondary information to understand the specific context and intent. Ensure the initial answer is logically consistent with the provided data by performing a preliminary check. Pay special attention to key phrases or terms that might indicate specific contexts or conditions, such as dates and ages, to enhance contextual understanding. Define what constitutes a "project" for each entity, specifying categories such as TV shows, movies, specials, and spinoffs, and apply these definitions consistently. Evaluate the temporal context by assessing the timeline of events and their relevance to the current year or the year in question. Cross-reference multiple reliable sources for fact verification, listing potential sources and checking for consistency before finalizing the answer. Use a secondary model or external knowledge base for confirmation when necessary. Ensure all information directly contributes to the conclusion, explicitly stating how each piece supports the boolean answer. Structure the explanation in a step-by-step manner, ensuring each point logically leads to the conclusion. Recognize and address any hypothetical scenarios or specific conditions by identifying keywords or phrases that indicate such situations. Break down the question into logical components and evaluate each against the given conditions to determine the boolean outcome. Consider potential edge cases and how they might affect the outcome. Use precise language and clarify any terms that could be interpreted in multiple ways. Incorporate strategies for handling ambiguous queries, such as identifying potential ambiguities, seeking clarification, or providing a probabilistic answer when certainty is not achievable. Implement a confidence scoring mechanism to express certainty in the answer, providing a probability score or a statement of uncertainty when not fully confident, prompting further verification or clarification. Be aware of common biases and heuristics, critically evaluating their applicability to the specific case and considering exceptions to general rules. Reflect on training data to recall previous similar questions and their resolutions, applying learned patterns to new queries. After formulating an initial response, verify the answer by cross-referencing with a reliable knowledge base to ensure accuracy. Perform a self-assessment by reflecting on potential errors or misinterpretations, and adjust the response accordingly. Ensure alignment with the ground truth by checking the conclusion against a known correct answer or reliable source. Provide a clear and robust justification for the answer, exploring both direct and indirect factors. Use definitive language to avoid ambiguity and ensure logical consistency throughout the explanation. Focus on the specific query, filtering out extraneous details and concentrating on elements crucial to answering the question accurately. Use the following refined example as a reference for structuring your response: "Given the conditions that [condition], the answer is [True/False] because [reasoning]." Incorporate an iterative feedback and learning loop to refine understanding and improve accuracy over time, analyzing incorrect answers to identify errors and adjust strategies accordingly."

## A.4 LLM USAGE

We utilized an LLM solely for the purpose of refining the prose and enhancing the clarity of this paper. The model was prompted to correct grammatical errors, improve sentence structure, and polish writing. All intellectual contributions, including the core ideas, experimental design, and analysis, are exclusively the work of the authors. The LLM's role was strictly limited to that of a writing aid and did not contribute to the scientific content of this research.

