# OpenReview forum: "Pick Your Textual Gradients"
_ICLR.cc/2026/Conference — ICLR 2026 Conference Withdrawn Submission_

### Official Review · Reviewer_PyXM · 2025-10-30

**Soundness:** 3
**Presentation:** 3
**Contribution:** 2
**Rating:** 6
**Confidence:** 3

**Summary:**

This paper tackles why automated prompt optimization is so unstable. The authors find two main points: getting feedback on correct answers creates junk gradients, and focusing only on hard cases leads to an overly complex prompt that fails on simple tasks (overfitting).

Their solution, STEVE, is a two-step filtering process. First, it only learns from the model's failures. Second, it uses a preservation set of easier examples to verify that any new prompt candidate doesn't hurt general performance. This makes the optimization process much more stable and effective.

**Strengths:**

1.  The paper identifies a real, practical problem. The instability it describes is a major pain point for anyone working on automated prompting.
2.  The two-part solution is simple and makes perfect sense. It smartly borrows established ideas like hard-negative mining and preventing catastrophic forgetting.
3.  The results are very convincing. Testing across diverse tasks and multiple powerful LLMs shows that the method is robust and generally applicable.

**Weaknesses:**

1. The set used for verification is based on what the initial prompt gets right. It does not get updated, so it might not protect the new, more complex capabilities the prompt learns during optimization.
2.  The stability comes at a cost. STEVE requires more LLM calls per iteration to generate and verify candidates. This practical trade-off could be discussed more explicitly.

**Questions:**

I was curious about the annealing strategy for the regularization parameter λ. The paper mentions it is increased at each step, making the optimization more conservative over time. Could you elaborate on the intuition behind this choice, as opposed to keeping it constant or decreasing it?

---

### Official Review · Reviewer_PA7f · 2025-10-30

**Soundness:** 2
**Presentation:** 2
**Contribution:** 2
**Rating:** 2
**Confidence:** 4

**Summary:**

This paper proposes STEVE, a framework to stabilize textual gradient-based prompt optimization. Specifically, STEVE includes two core mechanisms: (1) Error-Driven Refinement, which ensures a high-quality signal by generating textual gradients exclusively from model failures; (2) Regularized Verification, which acts as a gate, accepting a candidate prompt only if its improvement on hard cases does not degrade performance on a fixed "preservation set". The authors validate its effectiveness on 10 complex instruction-following and reasoning benchmarks.

**Strengths:**

1. The idea of Regularized Verification is intuitive and sound. Tackling the trade-off between generalization and improvement on hard cases with a regularization objective ($Improvement - \lambda \cdot Regression$) is a straightforward and effective method.

2. The authors validate the proposed method across a wide range of 10 tasks with three different optimizer models, validating the effectiveness.

**Weaknesses:**

1. The core claims of this paper are based on weak experiments and unrigorous deduction, especially the claim of "noisy gradients from correct examples":
(1) the primary evidence comes from the w/ Correct-Only ablation in Table 2. However, the observed performance drops (<2%) are extremely small and only based on an average of 3 runs, which is well within the margin of random fluctuation for LLM prompting or even prompt rephrasing [1,2].
(2) the result are not supportive to deduce the claim that "producing feedback with noisy gradients that acts as a random perturbation". This claim is unrigorous without any direct evidence.


2. The two core mechanisms of the proposed method seems only re-brand standard components present in prior work. (1) "Generating feedback only from error cases" is a widely-adopted mechanism, which has been used in ProTeGi (Pryzant et al., 2023) and many of the following work. (2) The idea of using a dev set to select the best prompt is also standard, as have used in ProTeGi, [1-4] and many other work. The only new idea is the use of a trade-off regularization objective for prompt selecting.



[1] Large Language Models are Human-Level Prompt Engineers

[2] Are Large Language Models Good Prompt Optimizers?

[3] Prompt Engineering a Prompt Engineer

**Questions:**

See Weakness.

---

### Official Review · Reviewer_aBt9 · 2025-11-02

**Soundness:** 2
**Presentation:** 3
**Contribution:** 2
**Rating:** 4
**Confidence:** 4

**Summary:**

This paper studies instability in automated prompt optimization methods that leverage textual gradients. The paper identifies two major issues: (1) noisy gradients generated from correctly handled examples, and (2) over-specialization on difficult cases leading to degraded generalization. To address this, they propose STEVE, which focuses gradient updates on failed examples while validating each new prompt on a held-out preservation set. Experiments across 10 reasoning benchmarks (e.g., GSM8K, StrategyQA, Object Counting) demonstrate improved stability and consistent performance gains compared to existing baselines such as TextGrad and REVOLVE.

**Strengths:**

- The paper tackles an important and timely problem in automated prompt optimization for LLMs, focusing on the stability of iterative textual gradient methods.

- The proposed approach, while simple, is conceptually clear and well-motivated: “picking” failed cases for feedback and verifying with a generalization-preservation gate is intuitive and practical.

- The presentation is generally clear, and the problem formulation is rigorous.

**Weaknesses:**

- The two proposed mechanisms, i.e., selecting failed cases and verifying updates on a held-out set, are conceptually related to known ideas such as active learning. While their combination is effective, the methodological contribution feels somewhat incremental relative to existing APO frameworks like TextGrad and REVOLVE.

- The experiments can be extended to a more representative benchmark to demonstrate its effectiveness. It would be valuable to test STEVE on broader and more challenging domains (e.g., MATH500, code generation, open-domain dialogue, or safety-sensitive evaluation) to support claims of general robustness.

- Some of the motivational findings (e.g., “noise from correct examples” or “over-specialization on hard cases”) remain mostly qualitative. More quantitative evidence (e.g., measuring gradient variance, evaluating degradation curves over iterations) would make the analysis more convincing.

**Questions:**

See weaknesses.

---

### Author Response · Authors · 2025-11-22
**Withdraw of Paper**

Dear Reviewers,

We would like to express our sincere gratitude for your time and effort in reviewing our paper. We have carefully considered the constructive feedback provided by all reviewers.

While we believe in the potential of our idea, we acknowledge the reviewers' concerns regarding the novelty behind the Error-Driven Refinement. We realize that addressing these issues comprehensively requires significant revisions and additional experiments that cannot be completed within the rebuttal period.

Therefore, we have decided to withdraw our paper from ICLR 2026 to focus on improving the work based on your valuable suggestions. We plan to resubmit a stronger version of this work to a future venue.

Thank you again for your professional insights.

Sincerely, The Authors

---

### Note · Authors · 2025-11-22

**Comment:**

Dear Reviewers,

We would like to express our sincere gratitude for your time and effort in reviewing our paper. We have carefully considered the constructive feedback provided by all reviewers.

While we believe in the potential of our idea, we acknowledge the reviewers' concerns regarding the novelty behind the Error-Driven Refinement. We realize that addressing these issues comprehensively requires significant revisions and additional experiments that cannot be completed within the rebuttal period.

Therefore, we have decided to withdraw our paper from ICLR 2026 to focus on improving the work based on your valuable suggestions. We plan to resubmit a stronger version of this work to a future venue.

Thank you again for your professional insights.

Sincerely, The Authors

**Withdrawal Confirmation:**

I have read and agree with the venue's withdrawal policy on behalf of myself and my co-authors.